# Automated Quantitative Image-Derived Input Function for the Estimation of Cerebral Blood Flow Using Oxygen-15-Labelled Water on a Long-Axial Field-of-View PET/CT Scanner

**DOI:** 10.3390/diagnostics14151590

**Published:** 2024-07-24

**Authors:** Thomas Lund Andersen, Flemming Littrup Andersen, Bryan Haddock, Sverre Rosenbaum, Henrik Bo Wiberg Larsson, Ian Law, Ulrich Lindberg

**Affiliations:** 1Department of Clinical Physiology and Nuclear Medicine, Copenhagen University Hospital-Rigshospitalet, 2100 Copenhagen, Denmark; flemming.andersen@regionh.dk (F.L.A.); bryan.haddock@regionh.dk (B.H.); henrik.bo.wiberg.larsson@regionh.dk (H.B.W.L.); ian.law@regionh.dk (I.L.); ulrich.lindberg@regionh.dk (U.L.); 2Department of Clinical Medicine, Faculty of Health and Medical Sciences, University of Copenhagen, 2200 Copenhagen, Denmark; 3Department of Neurology, Copenhagen University Hospital, Bispebjerg, 2400 Copenhagen, Denmark; sverre.rosenbaum@regionh.dk; 4Functional Imaging Unit, Department of Clinical Physiology and Nuclear Medicine, Copenhagen University Hospital-Rigshospitalet, 2600 Copenhagen, Denmark

**Keywords:** image-derived input function, kinetic modelling, arterial input function, perfusion, positron emission tomography, long-axial field-of-view scanner, PET/CT

## Abstract

The accurate estimation of the tracer arterial blood concentration is crucial for reliable quantitative kinetic analysis in PET. In the current work, we demonstrate the automatic extraction of an image-derived input function (IDIF) from a CT AI-based aorta segmentation subsequently resliced to a dynamic PET series acquired on a Siemens Vision Quadra long-axial field of view scanner in 10 human subjects scanned with [^15^O]H_2_O. We demonstrate that the extracted IDIF is quantitative and in excellent agreement with a delay- and dispersion-corrected sampled arterial input function (AIF). Perfusion maps in the brain are calculated and compared from the IDIF and AIF, respectively, showed a high degree of correlation. The results demonstrate the possibility of defining a quantitatively correct IDIF compared with AIFs from the new-generation high-sensitivity and high-time-resolution long-axial field-of-view PET/CT scanners.

## 1. Introduction

Positron emission tomography (PET) is a powerful imaging modality that enables the quantification of radiotracer pharmacokinetics in living organisms [1]. PET plays a pivotal role in a wide range of biomedical research and clinical applications, providing valuable insights into physiological and pathological processes at the molecular level [2]. The quantitative pharmacokinetic analysis of time-resolved PET data allows for the determination of various physiological parameters, such as distribution volume(s), binding potentials, perfusion values, and/or metabolic rate constants [3,4,5,6]. Central to pharmacokinetic analysis is the accurate estimation of the arterial blood input function (AIF), which characterizes the time course of tracer delivery through the circulatory system to the tissue of interest.

Conventionally and historically, in the context of pharmacokinetic analysis by PET, the AIF has been obtained through invasive arterial blood sampling by cannulation, typically in a radial artery. Here, discrete arterial blood samples are withdrawn at regular intervals and subsequently measured with separate analysis equipment or measured continuously by dedicated samplers to determine the whole-blood radioactivity concentration as a function of time. This approach has long been considered the gold standard and has been widely employed for research purposes. However, it has several time-consuming and costly limitations preventing its widespread use in the clinic. These limitations stem from the invasive nature of the procedure, requiring specialised equipment and personnel with skills in the cannulation procedure to reduce the risk of complications. For many patients, the procedure can be complicated by difficult arterial access, patient consent, risk of clotting, and the constraints of the total extracted blood volume of the subject when frequent or repeated measurements are required [7]. Furthermore, it cannot be performed in patients receiving anticoagulant therapy and may have limited use in various patient populations such as children and patients with neurocognitive disabilities. Moreover, it requires technical equipment calibrated and cross-calibrated to the activity measurement and internal clocks of both the scanner and blood sampler. The blood sampling equipment itself may be impractical or unfeasible, particularly in the case of whole-body imaging, due to limited physical access to the subject. Finally, the peripherally sampled AIF needs postprocessing to correct for the effects of internal and external dispersion and differences in delay to accurately replicate the organ-specific AIF, which may induce inaccuracies [8]. We will, in this context, consider dispersion as the smearing out of the radiotracer bolus due to inhomogeneous velocity fields in vessels and external tubing, approximated in the current work as a monoexponential function, cf. Equation (2). The measured AIF can, hence, be described as the true AIF convolved with a monoexponential function with an appropriate time constant.

To overcome these limitations, the investigation of the use of image-derived input functions (IDIFs), i.e., obtaining the arterial tracer concentration directly from reconstructed time-framed PET images, has gained considerable interest [9,10,11,12,13,14,15,16]. The use of IDIFs, however, presents several challenges to standard short-axial field-of-view (SAFOV) clinical PET/CTs. These challenges include but are not restricted to the low achievable temporal resolution ultimately limited by the resulting signal-to-noise ratio in a single time frame that may induce scatter correction failure. A typical PET/CT SAFOV of 20–25 cm, covering only a narrow section of the patient at a time, limits studies outside the thoracic region to smaller arteries, complicating quantitative arterial IDIFs. Partial volume effects (PVEs) due to the limited spatial resolution of reconstructed PET images compared with the size of vascular structures lead both to activity spill-out and spill-in from neighbouring structures. These points lead to systematic absolute quantification SAFOV IDIF errors relative to the AIF standard of truth, ultimately affecting pharmacokinetic calculations and analysis, hence introducing both bias and variability.

The recent introduction of long-axial field-of-view (LAFOV) clinical PET/CT scanners with a simultaneous axial PET acquisition of 1 m or more has not only dramatically increased body coverage but also the sensitivity of up to 10 times the value of SAFOV PET/CT scanners due to the increased angular acceptance of coincidence events [17]. This technical advance directly addresses the known caveats of IDIFs, enabling a high temporal resolution of a second or less—for some implementations down to 100 ms [18]—at higher image signal-to-noise ratios while simultaneously maintaining whole-body scan coverage of major vessels and all vital organs. This enables the direct and non-invasive sampling of blood activity concentration in the central regions of large-diameter principal arteries, obviating the need for PVE correction and providing a self-containing dataset measured on the same equipment with identical calibrations and corrections.

The utilisation of IDIFs offers numerous advantages, including reduced invasiveness for the patient, a self-contained dataset with no need for additional decay correction and/or absolute time adjustment, improved patient comfort, and the ability to obtain input function measurements in situations where arterial sampling is not feasible. Moreover, high-temporal-resolution IDIFs provide more detailed information, allowing for the investigation of the regional heterogeneity of blood supply arrival due to, e.g., arterial steno-occlusive disease, ultimately affecting regional tracer kinetics and physiological quantitation, which may be missed when relying solely on a single whole-organ mean delay.

Here, we present a method for the automated extraction of an arterial IDIF extraction on a high-sensitivity Siemens Quadra LAFOV PET/CT scanner using [^15^O]H_2_O. [^15^O]H_2_O PET is considered the gold standard for in vivo tissue perfusion measurements through kinetic compartmental modelling and is heavily dependent on an accurate peak concentration determination of the input function to avoid biased parameter estimates. The derived IDIF was compared with an AIF measured simultaneously from arterial blood and compared directly between IDIF and AIF for the accuracy of modelling regional cerebral blood flow (rCBF) estimated by compartment modelling.

## 2. Materials and Methods

### 2.1. Phantom Data

Prior to the clinical studies, the recovery coefficient of the scanner using [^15^O]H_2_O was estimated from a custom-built phantom. An activity of 400 MBq of [^15^O]H_2_O was diluted with 2 L of water to a concentration of 200 kBq/mL, from which standard clinical plastic syringes (Omnifix, B. Braun (Ashbourne, Ireland)) with diameters of 8.7 mm, 12.0 mm, 14.6 mm, 19.1 mm, and 26.5 mm were filled. The syringes were submerged in a plastic box containing a background solution of 30 MBq [^15^O]H_2_O dissolved in 4 L of water (7.5 kBq/mL) used as a reference solution, resulting in an approximate target-to-background contrast of 27:1.The plastic box was centred in the PET bore, and scanning commenced covering a timeframe of 80–110 s after mixing the reference solution. The acquired data were reconstructed using a 3D-OP-OSEM with 4 iterations and 5 subsets using a maximum ring difference (MRD) of 85, corresponding to an axial angular coverage of 18°, including time-of-flight (TOF) information and point-spread function (PSF) correction in a 440 × 440 × 645 matrix resulting in 1.65 × 1.65 × 1.65 mm^3^ voxels. The reconstructed images were thereupon filtered with 0 mm, 2 mm, 4 mm, and 6 mm full-width half-maximum (FWHM) Gaussian filters, providing, in total, 4 image sets for each syringe diameter.

Circular volumes of interest (VOIs) were created with a diameter of 6 mm centred on the cross-section of 18 axial slices along the long axis (3 cm) of the respective syringes, with an additional VOI covering the reference solution. Activity recovery was calculated as the ratio of the mean activity of each VOI in the syringes divided by the average of the reference solution.

### 2.2. Human Data

All patients included had cerebrovascular steno-occlusive disease and were referred clinically for the measurement of the cerebrovascular reserve capacity. After inclusion, written and oral consent in accordance with ethical guidelines was collected. The study was approved by the departmental review board (Ref. no 481_21). In Cohort 1, ten patients (age: 32–79 years) underwent rest and acetazolamide (Diamox^®^ (London, UK)) scans, totalling 20 scan sessions with a hand-injected bolus (<5 s) of 400 ± 50 MBq [^15^O]H_2_O in a line connected to the right median cubital vein. The acetazolamide dose was 15–20 mg/kg administered intravenously over a 5 min period, and post-acetazolamide measurements were initiated 20 min later. Acetazolamide is a vasodilatory compound that increases rCBF by 30–60% [19,20,21] and is used clinically to estimate the cerebrovascular reserve capacity, which is an important predictor of ischemic events and is a prognostic factor used in work-ups prior to revascularising surgery. Arterial cannulation was performed after local anaesthesia in the non-dominant radial artery, and blood was sampled continuously at a frequency of 1 Hz with a scanner-synchronised clock and an activity cross-calibrated autosampler (Allogg Technology, Mariefred, Sweden), counting coincidences in a lead-shielded detector. The tube diameter was 1 mm, and the distance from cannulation to sampler was 1.5–2.0 m. The blood withdrawal rate was 4 mL/min.

PET data were acquired on a Siemens Vision Quadra LAFOV PET/CT scanner (Siemens Healthineers, Knoxville, TN, USA) with an axial FOV of 106 cm. To ensure optimal scanner sensitivity at an MRD of 85, the subjects were positioned fixed in a head holder approximately 15 cm from the end of the FOV, providing an effective subject coverage of approximately 90 cm. A dynamic PET acquisition was acquired for 4 min and initiated 10 s before the injection of [^15^O]H_2_O to establish a baseline. The dynamic data were subsequently reframed in 40 × 1 s, 5 × 4 s, 6 × 10 s, 3 × 20 s, and 2 × 30 s frames totalling 4 min and reconstructed for the phantom data using 3D-OP-OSEM with 4 iterations and 5 subsets, including TOF and PSF correction (TrueX) in a 440 × 440 × 645 matrix, resulting in 1.65 × 1.65 × 1.65 mm^3^ voxels. The reconstructed data were filtered by a 3D Gaussian filter with an FWHM of 2 mm. PET data were attenuation-corrected in-loop with a low-dose CT scan matching the rest or acetazolamide scans.

The aorta was segmented on CT scans matched to each respective scan phase by the TotalSegmentator v2.02 algorithm [22] based on a nnU-Net [23]. The entire thoracic aorta length was divided into four segments, namely, the ascending, upper proximal descending, upper distal descending, and lower descending part. The aorta mask defined by TotalSegmentator was continuously pixel-wise eroded from the rim towards the centre to a total volume of approximately 1 mL. This volume size definition was deemed optimal for the IDIF acquisition with regard to the area under the input function curve (AUC) while simultaneously keeping the pixel noise sufficiently low. Code for the automatic IDIF extraction given an aorta mask and a dynamic PET series is available at https://github.com/Rigshospitalet-KFNM/IDIF.

The AIF was decay-corrected at the start of the PET acquisition. Both the IDIF and the AIF were delay- and dispersion-corrected by simultaneous fitting, *K*_1_; the volume of distribution, *V_d_*; delay, *Δt* (s); and dispersion, τ (s), in a one-tissue compartment model according to the method proposed by Meyer [24], Equation (1), to the mean tissue time activity curve (TAC) from 0 to 3 min post-scan start, calculated within the TotalSegmentator-inferred brain mask.
(1)$$C_{t}\left(t\right)=\tau K_{1}C_{a}\left(t-∆t\right)+\left(1-\tau \frac{K_{1}}{V_{d}}\right)K_{1}C_{a}\left(t-∆t\right)⨂e^{-t\cdot \frac{K_{1}}{V_{d}}}$$

*K*_1_ represents the perfusion under the assumption of the full extraction of the radiotracer, τ the dispersion, and *Δt* the delay. The fitted parameters for the application of both the IDIF and AIF to the mean brain TAC, namely, the dispersion and delay, were subsequently convolved according to Equation (2).
(2)$$d\left(t\right)=\frac{1}{\tau }exp\left(-\frac{t}{\tau }\right)$$

Equation (2), hence, represents a simple monoexponential dispersion function, as previously employed [8]. The convolution of the IDIF was preferred as opposed to the deconvolution of the AIF to reduce noise and deconvolution artefacts in the input functions.

For test–retest variability and to test the robustness of the IDIF, an additional 30 patients (Cohort 2, age: 16–83 years) were included in the study and underwent two rest and two acetazolamide scans. No arterial sampling was performed on this cohort. For analysis of these data, a one-tissue compartment model without dispersion was fitted to estimate the perfusion; distribution volume, *V_d_*; and input function delay relative to the tissue.

For both cohorts, the global absolute and relative cerebrovascular reserve (%) was calculated.

Within-subject reproducibility was estimated in terms of the within-subject coefficient of variation (COV) using a logarithmic method [25], as was the repeatability or absolute intra-subject variability, calculated as (test–retest)/mean(test and retest) × 100, and the reliability (intraclass correlation coefficient, ICC) as a two-way, random, single score.

The ICC represents the proportion of the variability not attributable to measurement error and was calculated as
(3)$$ICC=\frac{MS_{S}-MS_{E}}{{MS}_{S}+\left(k-1\right)MS_{E}+k\left(MS_{T}-MS_{E}\right)/n}$$
where *MS_S_* is the subject mean square, *MS_E_* is the error mean square, *k* is the number of trials, *MS_T_* is the trial mean square, and *n* is the number of subjects. An *ICC* > 0.9 is considered excellent and the lowest acceptable standard for measurements from which diagnostic decisions can be made.

## 3. Results

### 3.1. Blood Input Function Comparisons

The AIF from external blood sampling and the IDIF from the volume in the aorta were used to model the mean time activity curve using Equation (1) co-fitting perfusion, *K*_1_; tissue distribution volume, *V_d_*; dispersion, *τ*; and delay between tissue and respective input function, *ΔT*. A representative result is shown in Figure 1.

Average arterial delays of 0.8 s (range: 0.1–1.7 s) and 0.38 s (range: 0.1–1.2 s) for the rest and acetazolamide phases were found, respectively, indicating the arrival of the tracer to the distal thoracic descending aorta VOI prior to the brain. The dispersion was, in all cases, negligible for the IDIF, with averages of −0.1 s (range: −0.65–0.0 s) and −0.04 s (range: −0.5–0.0) for the rest and acetazolamide phases, respectively. In contrast, the dispersion values obtained for the AIF were, as expected, much higher, cf. Figure 1, with average dispersions of 8.4 s (range: 6.1–12.6 s) and 10.3 s (range: 8.1–13.2 s) for the rest and acetazolamide phases, respectively. The fitted delay of the of the AIF were −11.5 s and −12.3 s for the rest and acetazolamide phases, respectively.

The resulting parameter estimations using the one-tissue compartment model resulted in no statistically significant differences in perfusion values, volume of distribution, or absolute relative cerebrovascular response across subjects, as evaluated by a paired *t*-test, cf. Figure 2.

Since dispersion, *τ*, was found to be negligible in Cohort 1, it was omitted from Equation (1) for the estimation of parameters for the test–retest analysis in Cohort 2. For the rest and acetazolamide scans, the within-condition intra-subject coefficients of variation (CoVs) were 4.2% and 5.1%, respectively. The ICC and absolute intra-subject variability were 0.97 and −0.1% (range: −25.0–19.2%) for the rest phase, while corresponding values for the acetazolamide phase were 0.96 and 0.0% (range: −13.5–16.0%), showing a high degree of agreement.

The rest-to-acetazolamide phase perfusion increase was, on average, +30.9% (±7.1%) using the AIF and 37.1% (±9.8%) using the IDIF, which was, however, not statistically significant (*p* = 0.07), cf. Figure 2.

The fitted dispersion values found for the AIF were used to convolve the delayed IDIF according to Equation (2) rather than to deconvolve the external AIF to reduce noise amplification and optimize the comparison. The result from a representative subject is shown in Figure 3. An excellent agreement AUC between the AIF and IDIF with means of 8188 kBq s/mL (±2278 kBq s/mL) and 8210 kBq s/mL (±2312 kBq s/mL) was found, corresponding to an average relative difference between the AIF and IDIF of −0.3%. Thus, the presented method provides a robust and quantitative IDIF directly comparable to an externally sampled AIF.

A regional perfusion parameter maps, *K*_1_, using the AIF and IDIF are shown in Figure 4, along with a subtraction map. The difference map shows a small absolute difference between the rest and acetazolamide conditions of below 2 mL/100 mL/min.

### 3.2. Aorta Segmentation

The segmentation of the aorta from the CT scan proved robust in the current cohort as per visual inspections of all cases. From the aorta subdivisions, Segment 1, i.e., the IDIF in the ascending aorta, was observed to be scatter-affected by the high-activity concentration bolus in the adjacent pulmonary artery, providing a small overestimation of blood activity in the first 10 s (Figure 5). Segments 2, 3, and 4 were affected only to a minor extent by this effect.

As expected, minor differences in peak tracer bolus arrival times were observed following the blood flow of the circulatory system, namely, the ascending part of the aorta, the aortic arch, the proximal part of the descending aorta, and the distal part of the descending aorta. The time difference between Segment 1 and Segment 4 was approximately 2 s.

Segment 4 was used universally across subjects, as this proved to be the most robustly affected by the lowest amount of cardiac-and-respiratory-induced motion and scatter from nearby high-activity concentration vascular volumes.

### 3.3. Phantom Data

The recovery coefficients decreased with decreasing syringe diameter, as predicted. The loss of recovery coefficient reflected the ratio of the VOI diameter with respect to the syringe diameter and was further reduced by an increased reconstruction filter width. Recovery coefficients were approximately 1 for the 26.5 mm diameter syringe regardless of the reconstruction filter width, as seen in Table 1 (see also Appendix A). The population average diameter of the descending aorta representing aorta segment 4 was 25 (female) to 27 (male) cm on CT [26]. The syringe with a diameter of 26.5 mm was considered the most representative of this segment, and here, a 2 to 4 mm postfiltered reconstruction proved to serve as a good compromise for securing clinical brain PET image quality and the accurate quantification of the IDIF without additional spill-in/spill-out corrections.

## 4. Discussion

IDIFs obtained with LAFOV PET/CT provide an excellent alternative to invasive blood sampling, as they enable the direct measurement of the AIF from PET image data alone. Several methods have been proposed to extract the IDIF, including the manual delineation of images, automated segmentation algorithms, and hybrid approaches combining manual and automated techniques [27].

For the current study, a central VOI in the distal part of the upper thoracic descending aorta (cf. Figure 1) using a high-resolution PET reconstruction was deemed the most robust for extracting the IDIFs, minimising both PVE, scatter from the pulmonary arteries and respiratory and cardiac motion. When imaging other isotopes with lower positron energy than ^15^O and slower pharmacokinetics than [^15^O]H_2_O, it is possible that IDIF sampling from alternative areas such as the upper ascending aorta or the left ventricle could be considered.

Negligible dispersion correction was necessary due to the elimination of external dispersion and an approximate match of the internal dispersion between the distal internal carotid arteries and the upper thoracic descending aorta. This is a great advantage of the proposed IDIF method, as the maximum values of the sensitivity function (SF)—i.e., the most significant part of the fit to the TAC, for delay, perfusion, and distribution volume—all overlap in the initial time frames of the TAC [28]. Additionally, these parameters are not mathematically orthogonal, providing a large interdependence of the parameters in the fit. Thus, by eliminating dispersion correction, a more robust estimation of rate constants, volume of distribution, and delay is obtained.

The appropriate selection of the VOI was defined as a fixed volume in the present study. However, an individualised selection of the VOI based on the signal-to-noise ratio (SNR) in the extracted IDIF could decrease the absolute volume in some patients, resulting in the potentially lower dispersion of the IDIF in the aorta flow direction. However, the dispersion difference between aortic segments was found to be negligible, suggesting that this is a minor effect, at least with respect to the brain. For other organs with more complex and/or smaller vascular supply, such an effect may become more significant. Alternatively, partial volume corrections for activity spin-in and spill-out can be applied if the geometry, i.e., the surface area and volume, of the feeding artery is known [29].

A great strength of IDIF extraction is the consistent measurement of blood and tissue activity concentration. Due to the nature of VOI extraction directly from images, time and activity cross-calibration issues and blood handling are avoided, creating a self-contained examination. As an example of this, in a patient excluded from the study, a scanner malfunction led to imprecise calibration. This, of course, poses a detrimental problem for the use of externally sampled AIFs, which rely on the accurate cross-calibration of the blood sampler and scanner, while an IDIF-based analysis would be immune to such an issue.

The application of the IDIF as an input for kinetic models holds great potential for increased patient compliance and convenience. In the current study, we have examined and compared [^15^O]H_2_O with AIF and IDIF extracted from whole blood and found excellent agreement. Due to the free diffusivity and rapid mixing of [^15^O]H_2_O within the blood, the tracer concentrations in the blood compartments are identical. For the application of this approach to other tracers, however, this is usually not the case, and other factors, such as plasma-to-whole-blood equilibration, corrections for metabolites, protein binding, etc., pose a challenge, and they must be considered for accurate modelling. The blood compartment can be approached heuristically using a whole blood-to-plasma ratio from the literature [30,31,32]. For the correction of metabolites, this poses a greater challenge, although, with the improved sensitivity and SNR from new generations of LAFOV PET/CT scanners, the inclusion of additional metabolite compartments (rate constants) may prove viable.

From the phantom experiments, an activity recovery of 1 was found for a 6 mm diameter VOI placed centrally in a cylindrical syringe with a diameter of 19.1 mm or above with a 2 mm Gaussian postfilter following reconstruction. This poses no challenge for the healthy human adult population, where the aorta typically is 20 mm or larger. However, in young adults and children, the aortic diameter is significantly smaller, reducing the recovery coefficient. Care should be taken in such cases to obtain a quantitatively correct input function.

The use of digital LAFOV PET/CT scanners with high time resolutions, spatial resolutions, and sensitivity enables the extraction of an IDIF without additional postprocessing, as otherwise employed in other published methods [9,33]. Indeed, typically, the IDIF extracted from high-time-resolution images shows a lower full-width half-maximum (FWHM) than the dispersion correction AIF, suggesting a native high-quality IDIF directly from these images. It should be noted that, owing to the LAFOV, additional AIF dispersion arose from a doubling of the usual tube distance from the arterial cannulation to the sampler to 1.5–2.0 m, which approximately doubled the dispersion values to the 6.1 to 13.1 sec range compared with what is expected from the standard SAFOV PET/CT set-up [34].

We found excellent same-day test–retest global perfusion measured in a separate cohort of 30 clinical patients with an absolute variability of 0.1% or below; a reliability, ICC, of 0.96 or above; and within-condition intra-subject global CBF CoVs of 4.2% and 5.1% for rest and acetazolamide, respectively. The test–retest CoV has previously been reported in more recent [^15^O]H_2_O studies in healthy subjects using an AIF of 5.7–8.8% [35,36] during rest using SAFOV PET scanners and 7.1% during rest (*n* = 18) and 7.3% post-acetazolamide (*n* = 19) using PET/MRI [33]. In a mixed group of healthy subjects (*n* = 12) and patients with steno-occlusive disease (*n* = 7), test–retest CoVs of 5.6% for rest and 12.6% for post-acetazolamide were found using PET/MRI and spill-in- and spill-out-corrected IDIFs from internal carotid arteries [9]. All in all, global CBF measurements based on IDIF from LAFOV PET/CT are as robust or better than values reported using the AIF.

## 5. Conclusions

An automatically derived IDIF from a LAFOV PET/CT scanner was evaluated and found to produce the same results in a clinical setting as an externally sampled arterial input function. We have demonstrated the feasibility and, in some cases, the superiority of the IDIF with no dispersion to the brain. The extracted IDIF was accurate in quantitative terms, as demonstrated by both phantom experiments and a comparison to an externally sampled AIF. The extraction of an IDIF to obtain unbiased pharmacokinetic parameters in LAFOV scanners is, hence, viable and can, in the future, enable robust CBF measurements, obviating patient trauma and the risk of arterial cannulation and reducing procedural complexities associated with arterial blood sampling.

The test–retest metrics using the IDIF were equal to or superior to previous AIF-based measurements, testifying to the suitability of the approach for clinical use.

## Figures and Tables

**Figure 1 diagnostics-14-01590-f001:**
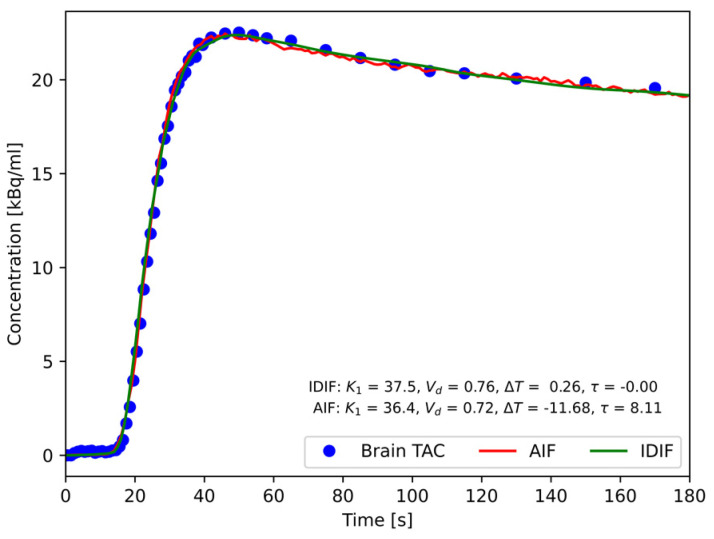
Representative fits to the mean brain TAC using Equation (1) for IDIF and AIF input functions. The resulting values of the unidirectional clearance of the tracer from blood to brain tissue, *K*_1_ (mL/(100 mL/min)); volume of distribution, *V_d_* (mL/mL); delay of input function from point of measurement in descending aorta (segment 4) and radial artery to brain arrival (Δ*T*); and dispersion (*τ*) are presented.

**Figure 2 diagnostics-14-01590-f002:**
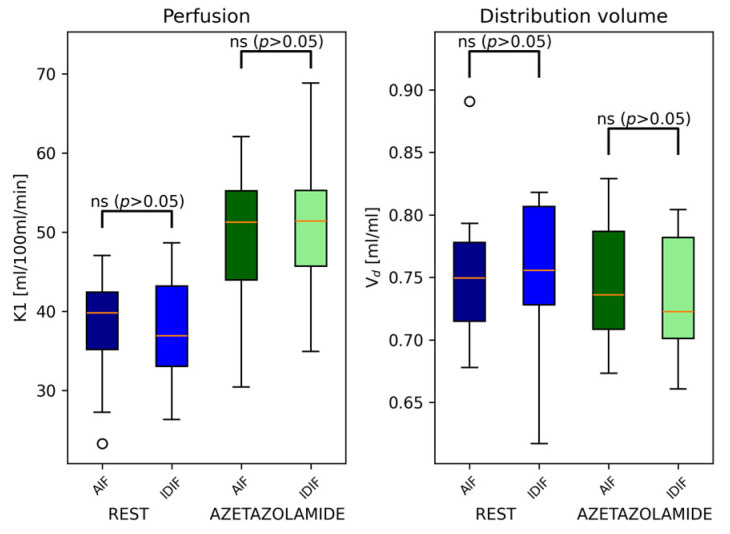
Comparison of the calculated perfusion, *K*_1_, and distribution volume, *V_d_*, using the AIF and IDIF alternatively as input functions in Cohort 1 (*n* = 10). No significant difference in obtained values was found between the two input functions. (o, standard notation for boxplots).

**Figure 3 diagnostics-14-01590-f003:**
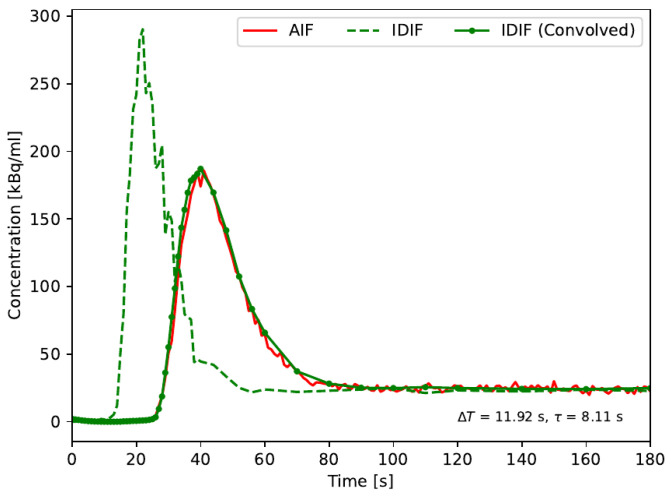
Direct comparison between the measured IDIF and AIF for a representative subject. The IDIF—is unaffected by tubing and measurement device transit times, unlike the AIF—was first delay-corrected to the AIF. Dispersion was subsequently added to the IDIF by convolution with the fitted time constant. In general, excellent agreement between the two input functions was found, with an average relative agreement of AUC of −0.3%.

**Figure 4 diagnostics-14-01590-f004:**
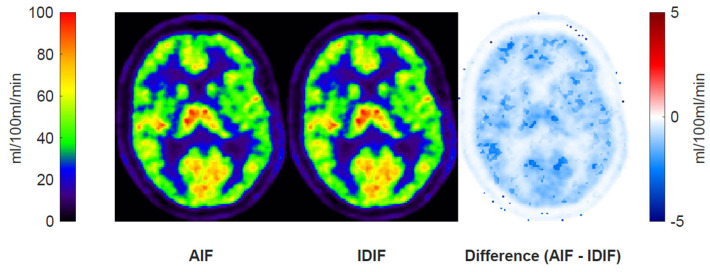
Perfusion (*K*_1_) maps and the absolute difference of these from a representative subject subjected to delay- and dispersion-corrected IDIF and AIF. Generally, very small differences in perfusion were observed.

**Figure 5 diagnostics-14-01590-f005:**
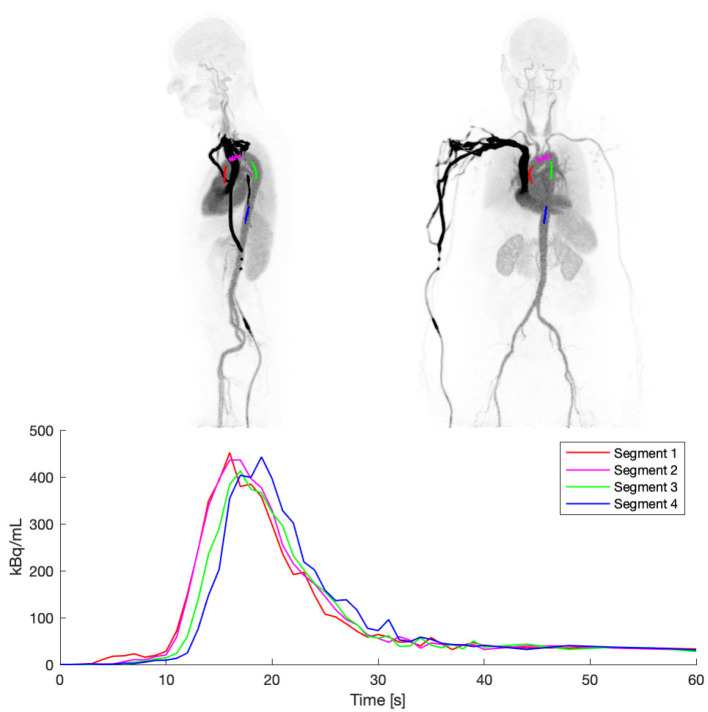
IDIF from the central part of CT-based segmentations of the thoracic aorta. Time course of the 4 different segments in the lower panel shows differences in delay and effects of uncorrected scatter, especially evident for Segment 1.

**Table 1 diagnostics-14-01590-t001:** Activity recovery coefficient as a function of syringe diameter and post-reconstruction Gaussian filters. Recovery is calculated as the ratio between the mean of a central 6 mm diameter VOI and the reference solution.

Diameter of Syringe/FWHM	0 mm	2 mm	4 mm	6 mm
26.5 mm	1.00	1.01	1.01	1.02
19.1 mm	1.01	1.01	1.00	0.97
14.6 mm	0.91	0.90	0.86	0.78
12.0 mm	0.83	0.80	0.73	0.63
8.7 mm	0.63	0.60	0.50	0.41

## Data Availability

The data supporting reported results can be obtained via contact with the corresponding author upon reasonable request and legal approval. The data are not publicly available due to no public data sharing agreement. The code for IDIF generation is available at https://github.com/Rigshospitalet-KFNM/IDIF.

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
