# Peer review of "Automated Quantitative Image-Derived Input Function for the Estimation of Cerebral Blood Flow Using Oxygen-15-Labelled Water on a Long-Axial Field-of-View PET/CT Scanner"

_diagnostics, 2024, doi:10.3390/diagnostics14151590_

Round 1
Reviewer 1 Report
Comments and Suggestions for Authors
Great study, well designed, executed, described and discussed.
My only minor comment is on Fig 2. "DIAMOX" is not mentioned as frequently in the text as "ACETAZOLAMIDE", if possible replace it in the label axis for more clarity.
Author Response
Comment: My only minor comment is on Fig 2. "DIAMOX" is not mentioned as frequently in the text as "ACETAZOLAMIDE", if possible replace it in the label axis for more clarity.
Answer: Thank you pointing this out. A very good suggestion for improvement. We have changed the label in the figure accordingly.
Reviewer 2 Report
Comments and Suggestions for Authors
This study aims to facilitate determination of arterial input function for cerebral blood flow determination. Here is proposed a non-invasive method, avoiding iterative arterial blood sampling.
Mathematical and physical considerations have been precisely taken into account. I particularly appreciate that “convolution of the IDIF was preferred as opposed to deconvolution of the AIF to reduce noise and deconvolution artefacts in the input functions”.
Minor comment are following.
General formal considerations :
No point after unit symbols !
What about the unit of MRD ? Degrees ?
Please, no * in units…
All mathematical and physical notations need to be checked (such as exponents, indices…).
Abstract :
« the arterial blood concentration » : please prefer “tracer arterial blood concentration”.
1. :
Line 59 : at this point, please define dispersion more precisely. Indeed, dispersion takes a major part in the discussion, namely internal and external.
2.1. :
Line 107 : “with 2 l. of water” ; prefer 2 L of water ; and so along the text.
2.2. :
Line 130 : “< 5 sec” ; please, 5 s !
4. :
Line 341 : usually instead of usual.
Author Response
Thank you for kind suggestions improved our manuscript. We have made the corrections to the units as suggested. See below for more detailed comments.
What about the unit of MRD ? Degrees ?
Answer: The unit of MRD is number of axial crystals. We realized that this is ambiguous and have added line 119-120 adding the axial angular converage: ", corresponding to an axial angular coverage of 18°,"
Abstract: « the arterial blood concentration » : please prefer “tracer arterial blood concentration”.
Answer: Thank you for the valuable suggestion. We have changed the wording accordingly.
Comment 1: Line 59 : at this point, please define dispersion more precisely. Indeed, dispersion takes a major part in the discussion, namely internal and external.
Answer: We agree that this is an important point. To clarify and improve the text we have added the following sentences to the text (lines 60-65): "We will in this context consider the dispersion as the smearing out of the radiotracer bolus due to inhomogeneous velocity fields in vessels and external tubing approximated in the current work as a monoexponential function, cf. Eq. 2. The measured AIF can hence be described as the true AIF convolved with a monoexponential function with appropriate an appropriate time constant."
Comment 2: Line 107 : “with 2 l. of water” ; prefer 2 L of water ; and so along the text.
Answer: We have changed the notation of the symbols as per request.
Comment 3: Line 130 : “< 5 sec” ; please, 5 s !
Answer: Changed as part of unit cleanup.
Comment 4: Line 341 : usually instead of usual.
Answer: Thank you for correction. Acknowledged and changed.
Reviewer 3 Report
Comments and Suggestions for Authors
This manuscript proposed an image derived input function in PET/CT scan for a non-invasive cerebral blood tracer. The results show that no significant difference between IDIF and AIF. The delay equation is utilized such that there is no delay time between IDIF and AIF.
1. Please elaborate in the manuscript why there is a delay time between IDIF and AIF in Figure 3.
Comments on the Quality of English Language-
Author Response
Thank you for the valuable comments. We have reread the manuscript and corrected typos and wordings to improve the language as suggested.
Comment 1: 1. Please elaborate in the manuscript why there is a delay time between IDIF and AIF in Figure 3.
Answer: Thank for the valuable point. We understand that the reason for the difference in delay here can be puzzling. Generally, the delay for arterial sampling is composed of not only blood transit time but also blood transport time in catheter, tubing and measurment device, The arterial delay needs correction for all of these parts. The IDIF, not having tubing and measurement transit time, is hence delay corrected to the AIF to facilitate direct comparison. The clarify this in the text we have changed to figure caption of Fig. 3 to: "Direct comparison between measured IDIF and AIF for a representative subject. The IDIF, not affected by tubing and measurement device transit times as opposed to the AIF, was firstly delay corrected to the AIF. Dispersion was subsequently added to the IDIF by convolution with the fitted time constant. In general excellent agreement between the two input functions was found with an average relative agreement of AUC of -0.3%."